# Effectiveness of Health Coaching in Diabetes Control and Lifestyle Improvement: A Randomized-Controlled Trial

**DOI:** 10.3390/nu13113878

**Published:** 2021-10-29

**Authors:** Ching-Ling Lin, Li-Chi Huang, Yao-Tsung Chang, Ruey-Yu Chen, Shwu-Huey Yang

**Affiliations:** 1Endocrinology & Metabolism, Cathay General Hospital, Taipei 106438, Taiwan; work5halfday@cgh.org.tw (C.-L.L.); kettyendo@gmail.com (L.-C.H.); 2School of Medicine, College of Medicine, Taipei Medical University, Taipei 110301, Taiwan; 3School of Public Health, Taipei Medical University, Taipei 110301, Taiwan; promiselove02@gmail.com; 4School of Nutrition and Health Sciences, Taipei Medical University, Taipei 110301, Taiwan; 5Nutrition Research Center, Taipei Medical University Hospital, Taipei 110301, Taiwan; 6Research Center of Geriatric Nutrition, College of Nutrition, Taipei Medical University, Taipei 110301, Taiwan

**Keywords:** health coaching, diabetes, healthy diet

## Abstract

Background: The study aimed to look into the effectiveness of a 6-month health coaching intervention for HbA1c and healthy diet in the treatment of patients with type 2 diabetes. Methods: The study was carried out via a two-armed, randomized controlled trial that included 114 diabetic patients at a medical center in Taiwan. During the 6-month period, the intervention group had health coaching and usual care for 6 months, and the control group had usual care only. The outcome variables were HbA1c level and healthy diet for follow-up measurement in the third and sixth month. Results: The study discovered a significant decrease in HbA1c and health diet improvement after the 6-month health coaching. Patients in the intervention group decreased their daily intake of whole grains, fruits, meats and protein, and fats and oils while increasing their vegetables intake. Conclusions: Health coaching may be conducive to the blood sugar control and healthy diet of patients with type 2 diabetes. Further study on health coaching with higher-quality evidence is needed.

## 1. Introduction

Diabetes is a chronic disease that causes numerous deaths and inflicting a heavy burden on medical systems worldwide. The World Health Organization (WHO) notes that with the global prevalence rate of diabetes among adults topping 8.5% now [1], the prevention and management of diabetes as well as control of its risk factors should be a priority for every country. In Taiwan, the prevalence rate of diabetes among population aged 18 and older stands at around 9.82% [2], ranking second place in health insurance outlay [3]. Therefore, Taiwan has established the “Diabetes Shared Care Network,” integrating the resources of medical institutions nationwide and boosting the efficiency of health insurance outlay for diabetes care [4]. Some free services are available at participating institutions for patients who join the network, including health education, blood sugar test, and foot and eye check. The network also oversees the quality of medical services and health insurance outlay of diabetes. However, most lifestyle-associated services provided by the network are related to health education rather than behavior counseling and health coaching, which is a far cry from the situation in many developed countries.

For many chronic diseases, empowering patients to undertake health management by themselves is of utmost importance. Studies have shown that health education alone is not enough to facilitate patients to have long-term behavior changes [5]. Patient empowerment involves social support, problem-solving skill, and active learning [6,7], rather than just knowledge education; hence, over the past years, health and wellness coaching has been catching on in many countries in order to improve patients’ self-management. Health coaching is a patient-centered approach to disease management that focuses on patients’ decision and their actual behaviors [8], and most of the health coaching studies have centered on one or more health behaviors and patients’ self-efficacy. Some meta-analysis studies have pointed out that health coaching intervention can help patients improve their HbA1c by about 0.3–0.6% within 4–6 months [9,10]. There have been only a few studies on the effect of diet contents on healthy diet behaviors [11], and without a detailed description of food and calorie intake [12,13], they fail to shed light on whether and how coaching is effective in improving patients’ diet habits.

It is necessary to further explore the effects of health coaching on changes in the diet composition of diabetic patients. For people with diabetes, there is no set of dietary methods that can be applied to all patients, yet health education and advice must be carried out according to their conditions and living habits. At present, for research on health education, there has been literature that uses various nutrients as outcome indicators to explore the changes in diet composition and to clarify the kind of behavioral counseling that has changed the patients’ diet content [14]. For health coaches, there should also be such research to better explain the influence of coaching on the dietary content of diabetic patients, such as increasing the intake of vegetables, reducing the intake of saturated fatty acids and refined carbohydrates, etc. Therefore, this study looks into coaching interventions for healthy diets, taking Taiwan’s current diabetes health education guidelines as a reference indicator and evaluating its practicability and effectiveness in Taiwan.

In the study, a randomized controlled trial was designed to evaluate the effectiveness of health coaching intervention on patients’ blood sugar management and healthy diet by a certified coach. The aim of this study was to enhance patients’ healthy diet and improve the indicators of diabetes, especially the value of HbA1c. Our hypothesis was that participants having the health coaching intervention would reduce their excess daily calorie intake, make healthier food choices, and keep their diets more in line with the dietary recommendations in diabetic patients.

## 2. Materials and Methods

The study involved a 6-month coaching intervention in a two-armed, randomized controlled trial approved by the Institutional Review Board (IRB) of Cathay General Hospital in Taiwan. The Trial registration number was “www.isrctn.com” (ID number: ISRCTN14167790, accessed date: 12 July 2019). It is based on a previous study of ours with a similar intervention arrangement but different study design [15]. It included two groups of subjects, one with monthly coaching intervention coupled with shared diabetes care and the other with shared diabetes care only.

Between October 2019 and February 2020, of the 131 patients who were screened from the hospital database and invited by two physicians, 120 subjects were enrolled in the study, resulting in a recruitment rate of 91.6%. In total, outcome measures were available for the 58 participants in the intervention group and 56 in the control group, and only 6 participants withdrew within 6 months. (Figure 1)

### 2.1. Recruitment and Randomization Setting

Participants were recruited among patients with diabetes treated at Cathay General Hospital in Taipei from October 2019 to February 2020, while data collection was from October 2019 to August 2020. The first author screened and selected prospective patients with type 2 diabetes mellitus from the hospital’s database, which was followed by an independent researcher randomly assigning them to the intervention group and the control group by using computer-generated random numbers. Then, two physicians recruited them individually during their regular outpatient visits. Patients in the intervention group were informed of the coaching program by a health coach, whereas the patients in the control group were informed of the pre-posted questionnaire survey. Therefore, the physicians, patients, and analysts were unaware of the participants’ allocation group, and participants in the control group did not know that this was an interventional trial. Subjects must be patients between 20 and 75 years old, with type 2 diabetes for at least one year, an HbA1c of 7.0% or greater for the past six months (poor control for diabetes), exhibiting no clinical depression or cognitive impairment.

### 2.2. Sample Size

Following our previous study [15], for detecting 0.68% between-group difference in HbA1c and standard deviation of 1.35 with a probability of a type-I error of 0.05 and a power of 80%, each group must have at least 50 participants. Given a 20% dropout rate, the size of each group was set at 60. Analyses were conducted using GPower 3.1 software for Windows [16].

### 2.3. Intervention

Patients in the intervention group had in-person coaching at the outset, which was followed by monthly telephone coaching for a total of six months. Coaching was provided on a one-on-one basis by a professional health coach. In the first session, the coach would ask each participant to set an HbA1c goal and an initial target for behavioral change; in addition to focusing on healthy diet, other behaviors related to diabetes self-management such as physical activity, medical adherence, and regular self-monitoring blood glucose (SMBG) can also be set. To be specific, the coach would assist the patients in analyzing their diet and comparing it with the recommendations of the Diabetes Dietary Guidelines [17] and then guide the patients to set feasible and attainable goals and action plans by themselves. Then, the coach would discuss with patients over an implementation schedule for action plans and check the progress via a monthly call while reinforcing participants’ values and responsibilities of self-management. In other words, the health coach empowers patients to actively change their behaviors through guiding and applying what they have learned in health education to bring about more specific and feasible changes in diabetes self-management and thereby effectively promoting blood sugar management.

The coach has a master’s degree in public health and received over 120 h of coach training before being certified as International Coach Federation’s (ICF) Associated Certified Coach (ACC). Patients in the intervention and control groups both received basic health education on diabetes by certified diabetes educators and normal care under the Diabetes Shared Care Network program at Cathay General Hospital. All participants could contact educators to ensure access to adequate educational resources.

### 2.4. Outcome Measures

The study had main outcome variables about HbA1c and health diet. HbA1c was measured via a blood test conducted during subjects’ outpatient visits, usually at an interval of three months. Healthy diet analysis comprised of two indicators, with one presenting the daily calorie intake and the other the nutrients in each serving. It was evaluated on the basis of the diet record of patients in the previous seven days. Based on the guidelines of the Taiwanese Association of Diabetes Educators [17], the diabetes educators and nutritionists adopted a carbohydrate counting method in dietary education and counseling. The guide states that each serving of carbohydrates contains 15 g of carbohydrate, each serving of protein contains 7 g of protein, and each serving of fat contains 5 g of oil. In addition, the guide provides a reference table for the substitution calculation of various foods. According to the recommendations of this guide, for a normal person, it is recommended to consume 1.5–4 servings of whole grains (22–60 g carbohydrates), 3–8 servings of eggs, beans, fish, and meat (21–56 g protein), 1.5–2 cups of dairy products (12–16 g protein), 3–7 servings of fats and oils (15–35 g fats), 3–5 servings of vegetables (about 100 g per serving), and 2–4 servings of fruits per day (about 100 g per serving) depending on age and weight. For diabetic patients, diabetes health educators and dietitians will suggest that they adjust their diets according to the patients’ conditions and blood sugar goals, such as reducing the intake of saturated fats (e.g., more fish and beans, less red meat) and refined starch products (e.g., rice, bread, noodles) and increasing vegetable intake per day.

### 2.5. Statistical Analysis

A Chi-square test or *t*-test was employed to assess differences in sociodemographic factors, health diet, and HbA1c between the two groups at baseline (Table 1). A paired *t*-test was used to assess differences in the HbA1c level, daily calorie intake, and diet of each group, and ANCOVA was used to gauge the difference between the two groups in 3-months and 6-months pre–post difference and was estimated as a basis for adjusting the baseline value of each variable only, since there was no baseline demographic difference between the two groups (Table 2). A repeated-measure ANOVA test was employed to find the mean difference of HbA1c between the two groups at baseline, the third month, and the sixth month (Figure 2, and the sphericity test *p* = 0.183). We also tested the Pearson correlation between the difference of HbA1c level, daily calorie intake, and diet.

Finally, we used multiple regression to test the correlations between coaching intervention, changes in eating behavior, and HbA1c within 6 months (Table 3). The independent variables are selected according to the analysis results of Table 1 and Table 2; only significant variables in the univariate analysis will be included in the regression analysis, and it seems that there is no obvious confounding factor that needs to be adjusted. We adjusted the baseline HbA1c to reduce the detection bias. Since vegetable intake is not significant in each model, it is not included in the table here. In addition, because daily calorie intake is highly correlated with whole grains intake and fats and oils intake, it is not classified as an independent variable. Hence, four models are used in an attempt to explore the possible intermediary relationships between health coaching intervention, healthy eating behavior changes, and HbA1c.

All analyses were performed according to the intention-to-treat principle, and all tests were analyzed at a 95% significance level (*p* < 0.05). Analyses were conducted using PASW 22.0 software for Windows (SPSS, Chicago, IL).

## 3. Results

Table 1 shows the demographic characteristics of the 114 participants: 49% were females, the mean age was 62.0 years (SD = 8.88), 35.1% had a bachelor’s degree or higher, the mean HbA1c was 8.24% (SD = 1.03), and average daily intake was 1329 Kcal (SD = 321.16). On average, they ate 8.47 servings of whole grains (SD = 3.34), 4.20 servings of meats and protein (SD = 1.75), 0.30 serving of milk and dairy products (SD = 0.60), 2.46 servings of vegetables (SD = 0.84), 0.99 serving of fruits (SD = 1.23), and 6.35 servings of fats and oils (SD = 2.31) per day. There was no significant difference between the two groups at baseline.

Between October 2019 and August 2020, when the coaching intervention was carried out, the intervention group witnessed a significant decrease of 0.62% (CI = 0.35 to 0.90, *p* < 0.01) in HbA1c, which was a far cry from the insignificant decrease of 0.14% (CI = −0.36 to 0.09, *p* = 0.228) for the control group (Table 2 and Figure 2). Coaching intervention in participants’ lifestyle caused a significant decrease of 207.16 Kcal calorie intake per day (CI = 148.17 to 266.16, *p* < 0.01) plus a decrease in 2.07 servings of whole grains (CI = 1.42 to 2.72, *p* < 001), 0.47 serving of meats and protein (CI = 0.07 to 0.87, *p* = 0.02), and 0.78 serving of fats and oils (CI = 0.34 to 1.22, *p* < 0.01), although intake of vegetables increased by 0.41 serving (CI = 0.20 to 0.63, *p* < 0.01). In comparison, for participants in the control group, general medical care and health education only caused a decrease of both 0.84 servings of daily whole grains intake (CI = 0.33 to 1.35, *p* < 0.01) and 74.38 Kcal of daily calorie intake (CI = 28.38–120.37, *p* < 0.01).

An ANCOVA test revealed that the two groups differed significantly in HbA1c at 3 and 6 months (3 months *p* = 0.026, 6 months *p* = 0.015), and repeated-measures ANOVA analysis also revealed that the intervention group had a significant decrease in HbA1c (*p* < 0.01). The daily calorie intake as well as whole grains, vegetables, and fats and oils intake also had similar results (Table 2, marked as ^a^ and ^b^). The correlation analysis showed that the decrease in intake of whole grains, fats and oils, and daily calorie intake was significantly related to the decrease in HbA1c within 6 months (*p* < 0.001), but the increase in vegetables intake was only significantly related to the decrease in A1c within 3 months (*p* = 0.007). The decrease in whole grains intake was also significantly related to the decrease in the intake of fats and oils (*p* = 0.029) and the increase in the intake of vegetables (*p* = 0.014).

Table 3 showed that the intervention of health coaching could lead to a reduction of HbA1c (*p* = 0.015) within 6 months in Model 1; however, after adding the variable of reduced intake of whole grains, this correlation became insignificant (*p* = 0.093). The reduction in fats and oils intake was also apparent to the reduction in HbA1c (*p* = 0.020), but the intervention of health coaching remained significant in this model (*p* = 0.042). The meats and protein intake (*p* = 0.116), milk and dairy intake (*p* = 0.797), vegetables intake (*p* = 0.602), and fruits intake (*p* = 0.202) did not significantly affect the original correlations. This meant that changes in some dietary behaviors would completely or partially mediate the intervention of health coaching and the blood sugar control. In addition, since the total calorie intake had high correlation with the whole grains intake (Pearson r = 0.752, *p*< 0.001) and the fats and oils intake (Pearson r = 0.596, *p*< 0.001), it could not add into regression at the same time. We had tested and replaced the whole grains intake and fats and oils intake with the total calorie intake, and it produced a similar result as model 4, indicating that baseline A1c and total calorie intake were significant, and R^2^ was 0.333.

## 4. Discussion

The study found that a 6-month health coaching intervention significantly improved type 2 diabetic patients’ HbA1c by 0.62% as well as their healthy diet behavior. On top of a previous study of ours, which found that intervention could improve HbA1c by 0.68% [15], this study, via a randomized controlled trial, proved further the effectiveness of intervention method among patients with diabetes in Taiwan, which had reached the threshold of clinical effectiveness of HbA1c (0.3–0.5%) according to some guidelines [18,19], and it also surpassed most health coaching/motivational interviewing studies [5,9].

The study also discovered that health coaching is effective in improving health diet, as it can significantly reduce daily calorie intake and food intake while increasing vegetables intake, which is in line with health education guidelines for diabetes. We found that the behavioral changes of whole grains and fats and oils intake would completely or partially mediate the effect of health coaching intervention on blood sugar control, which could indicate the causal relationship between the three. In addition, according to the results of the correlation analysis, it seemed that the participants had increased their intake of vegetables, replacing the original excess intake of whole grains and fats. This may make their diet more in line with the recommendations of the Diabetes Dietary Guidelines [17].

Up to now, most studies in this field have focused on the effect of behavior intervention on patients with type 2 diabetes in healthy diet [20,21], but only a few have described specific diet habits, e.g., increasing vegetables and fruits intake [11,22], decreasing carbohydrate intake [23], or decreasing fats and oils intake [24,25]. For now, the major challenge for diabetes self-management is how to implement diet regimen faithfully or how patients can meet these dietary standards in a convenient manner without affecting their daily life and work, so that the diet change can become a lasting one. The goal of coaching is to induce a lasting change in subjects’ behavior and habit, so it is very important to discuss specific changes in diet contents. In Taiwan, diabetes educators and nutritionists invariably teach diabetic patients about how to calculate carbohydrate intake, even suggesting them to eat more “low glycemic index” food, e.g., eat whole grains and whole foods in place of refined starches, eat more vegetables and intake less saturated fat and sugar [17]. For studies on behavior coaching, it is very important to understand specific principles and specific results of dietary intervention. Therefore, future studies on diet education and coaching on patients with diabetes should be able to provide specific results of intervention and discuss the changes in and effect on diet contents instead of only mentioning a diet counseling method and scale scores.

In this study, we did not include diabetic patients with good blood sugar control (HbA1c between 6.5 and 6.9). This does not mean that this group of patients is not suitable for or does not need the counseling of health coaches, or rather, it is based on the consideration for research comparability. At present, most studies on diabetes-related behavioral counseling, such as health coaches or motivational interviews, are aimed at patients with HbA1c above 7.0%, with some even aiming at patients having HbA1c of 8.5% or 10% [5,26]. The reason may be that patients with poor diabetes control during the pre-test are more likely to have a more significant intervention effect. There are even studies asserting that diabetes health education for self-management behavior is only effective for those with HbA1c above 8%, making the self-management intervention effects less significant if the main research population is the one with better blood sugar control [27]. We believe that whether coaching can also help patients maintain better blood sugar control or further improve their blood sugar level to less than 6.5% may be a topic worth discussing in the future, and there is currently no such research for reference.

Coaching intervention is highly effective in helping patients form a healthy lifestyle, as it is more patient-oriented and focuses on actual behaviors more than traditional counseling or health education, which may explain the significant difference between the intervention group and the control group in the dietary habit in follow-up test. It appears that health education is insufficient in changing patients’ behaviors and habits [5,28]. This is the reason why health coaching is more effective than health education for behavioral change and habit building, which is a rarely mentioned adult learning theory [29]. Since adults’ learning pattern is based on experience and they tend to build a new behavioral pattern via adjusting their old experience, learner-centered and engagement-based learning is more effective than health education or counseling [30]. This does not mean that health education is less important, as it can help patients have correct and sufficient health knowledge, which is conducive to health coaching.

Based on our findings, the study proposes some suggestions to future studies and our medical systems. First, study on the effect of coaching intervention on dietary change should describe specific changes in food intake and total calorie intake after the intervention. It will be helpful knowing how coaching intervention improves patients’ blood sugar control via adjustment of dietary habit, which can also facilitate future studies and promote effective practices. Second, more studies on health coaching with high-quality evidence are needed. Until now, there are only a few coaching-intervention studies with randomized controlled trials, and most of them also lack information about coaching training, professional background, competency standard, credible supervision, and certification standards. Finally, for future applications in Taiwan, it is suggested that health coaching services should be conducted to improve medical-care quality and chronic disease prevention and management, with more tests done in various medical divisions.

The study has several strengths. First, an experienced health coach with ICF credentials was employed, enabling a high-quality coaching intervention, which is a far cry from the practice of the vast majority of studies in the field, which do not have certified coaches for intervention. Not only can the coach’s competence affect the effectiveness of coaching intervention in such studies, but the supervision of health coaching by professional institutions can assure its quality. Second, the study provides an in-depth analysis and description of patients’ dietary habits, which is absent in most other studies, helping other health coaching researchers design a more effective coaching intervention for healthy dietary behaviors.

The study has a number of limitations. First, the study was carried out in a period when COVID-19 was ravaging the world and people’s daily life was affected seriously by related restrictions, including those in Taiwan, despite its remarkable success in containing the pandemic [31]. The situation may affect the results of the study. Therefore, it is suggested that similar studies be carried out again after the pandemic ends to confirm the validity of the study’s findings during a normal time. Second, given the different eating habits of different races, cultures, and regions, the dietary counseling methods and analytical results of the study may not be directly applicable to Western countries. This is why we suggest that related studies have to describe more specifically the changes in eating behaviors. Third, in this study, HbA1c was used as the main outcome variable, whereas other blood biochemical indicators such as HDL and LDL were not included, making the discussion of the effects on dietary changes more limited. The reason why it is more conservative to only use HbA1c as the result indicator is mainly because this indicator is a more direct and stable indicator for diabetes blood glucose control compared to indicators such as fasting blood glucose, HDL, and LDL. Although the inclusion of other biochemical indicators may make the research results clearer, we believe that similar research designs should be used to explore other indicators in the future to test the effectiveness of health coaching for indicators other than blood sugar. Finally, since current diets and health education guidelines for diabetic patients in Taiwan do not differentiate and calculate fats and oils according to the types of fatty acids (such as saturated, monounsaturated and polyunsaturated fatty acids, etc.) but only make a rough distinction (such as nuts, seeds, red meat, white meat, fish, etc.), the analysis did not explore the impact of different types of oils on HbA1c, which may affect the effectiveness of the research. Therefore, we suggest that this limitation can be taken into consideration when conducting research related to health coaching in the future to further explore whether health coaches can counsel patients to change the intake of different types of fats and explore its impact on blood sugar.

## 5. Conclusions

The study finds that health coaching may be an effective way to improve the HbA1c and dietary behavior of patients with type 2 diabetes. However, more quality health coaching studies are needed in Taiwan, and the introduction of health coaching to Taiwan’s medical system is quite important. In addition, the integration of a health coaching service into Taiwan’s medical system is also suggested.

## Figures and Tables

**Figure 1 nutrients-13-03878-f001:**
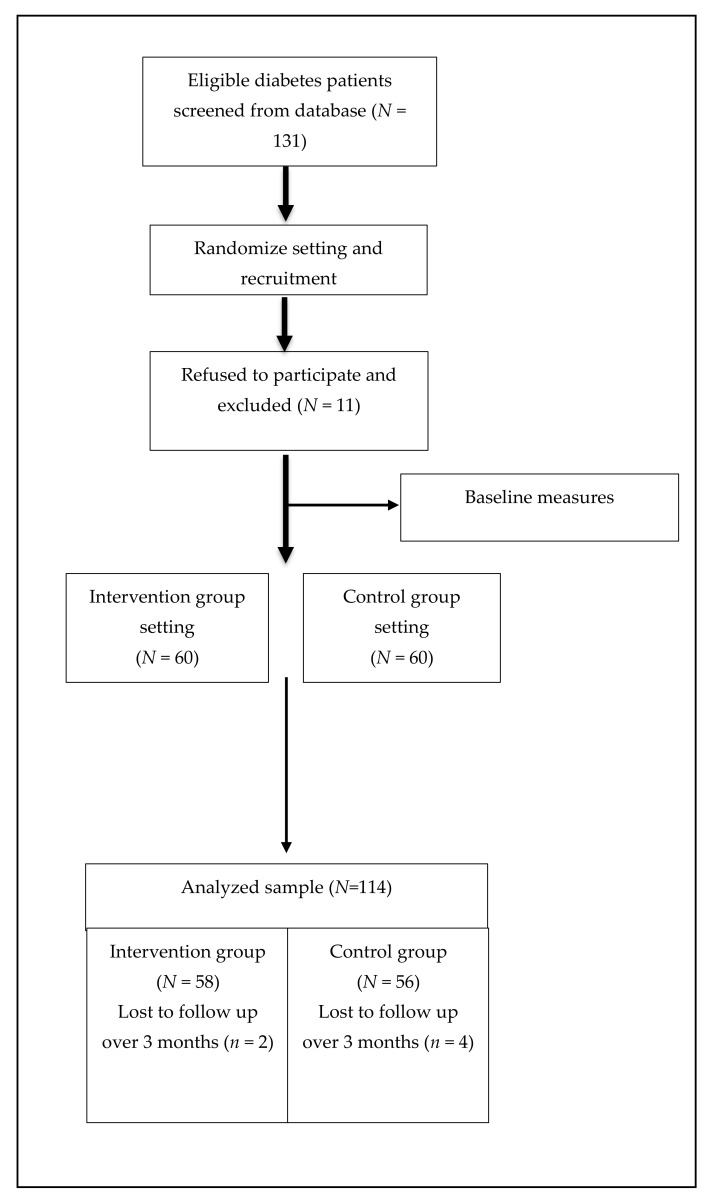
Flow diagram of participants: recruitment, intervention and follow-up.

**Figure 2 nutrients-13-03878-f002:**
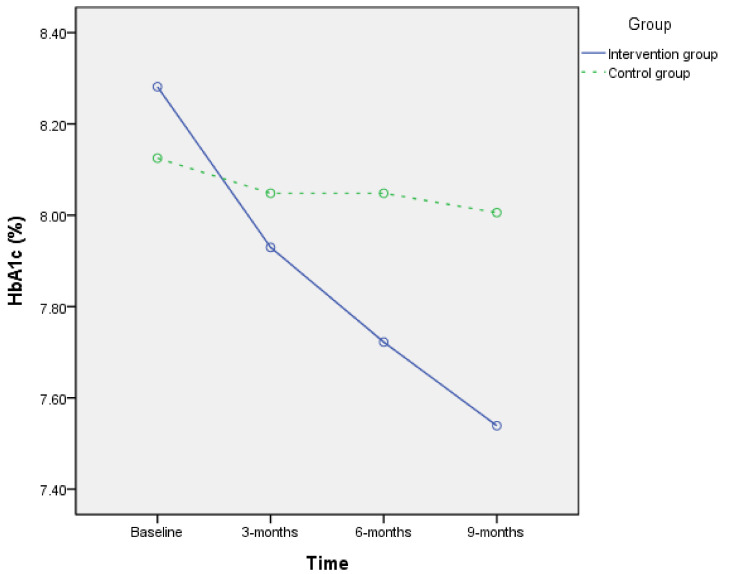
Effect of intervention within 9 months on HbA1c. Intervention group: solid line, control group: dashed line.

**Table 1 nutrients-13-03878-t001:** Demographic characteristics and baseline value of study groups.

	Demographic Characteristics *n* (%)		*p* Value
	Intervention Group (*n* = 58)	Control Group (*n* = 56)	
Gender			0.854
Male	30(51.7)	28(50.0)	
Female	28(48.3)	28(50.0)	
Age (years) ^a^			0.175
30–39	1(1.7)	0(0.0)	
40–49	10(17.2)	3(5.5)	
50–64	24(41.4)	26(47.3)	
65–75	23(39.7)	26(47.3)	
Educational level			0.133
Junior high school and bellow	10(17.2)	16(28.6)	
Senior high school	22(37.9)	26(46.4)	
University	19(32.8)	9(16.1)	
Master’s degree or above	7(12.1)	5(8.9)	
	Baseline Measure (mean ± SD)		*p* Value
HbA1c (%)	8.35 ± 1.19	8.13 ± 0.84	0.255
Daily calorie intake (Kcal)	1353.77 ± 332.12	1303.93 ± 310.35	0.410
Nutrition (serving/day) ^b^			
Whole grains	8.58 ± 3.32	8.36 ± 3.38	0.720
Meats and protein	4.46 ± 1.79	3.94 ± 1.68	0.113
Milk and dairy products	0.20 ± 0.46	0.40 ± 0.70	0.169
Vegetables	2.53 ± 0.84	2.38 ± 0.84	0.368
Fruits	1.03 ± 1.14	0.95 ± 1.32	0.747
Fats and oils	6.50 ± 2.45	6.19 ± 2.17	0.473

^a^ Participants aged 20–29 have not been recruited in our sample; ^b^ Based on 2019 Taiwan dietary guidelines.

**Table 2 nutrients-13-03878-t002:** Effectiveness of coaching intervention of study groups according to ANCOVA test and repeated-measured ANOVA test.

	Effectiveness of Coaching Intervention (Mean ± SD)		*p*-Value
	Intervention Group (*n* = 58)	Control Group (*n* = 56)	
HbA1c (%)			
Baseline	8.35 ± 1.19	8.13 ± 0.84	0.255
3 months	7.95 ± 1.09 ^a^	8.11 ± 0.99	0.417
6 months	7.73 ± 0.97 ^a,b^	7.99 ± 0.99	0.152
Daily calorie intake (Kcal)			
Baseline	1353.77 ± 332.12	1303.93 ± 310.35	0.410
3 months	1176.86 ± 275.91 ^a^	1248.35 ± 307.46	0.194
6 months	1146.60 ± 240.06 ^a,b^	1229.55 ± 283.17	0.094
Nutrition (serving/day)			
Whole grains			
Baseline	8.58 ± 3.32	8.36 ± 3.38	0.720
3 months	6.94 ± 3.10 ^a^	7.75 ± 3.07	0.164
6 months	6.51 ± 2.64 ^a,b^	7.52 ± 2.77	0.049 *
Meats and protein			
Baseline	4.46 ± 1.79	3.94 ± 1.68	0.113
3 months	4.09 ± 1.53	3.88 ± 1.54	0.464
6 months	3.98 ± 1.53	3.89 ± 1.54	0.755
Milk and dairy products			
Baseline	0.20 ± 0.46	0.40 ± 0.70	0.069
3 months	0.17 ± 0.43	0.41 ± 0.70	0.031 *
6 months	0.27 ± 0.49	0.47 ± 0.79	0.095
Vegetables			
Baseline	2.53 ± 0.84	2.38 ± 0.84	0.368
3 months	2.67 ± 0.80	2.38 ± 0.83	0.054
6 months	2.94 ± 0.85 ^a,b^	2.39 ± 0.81	<0.001 **
Fruits			
Baseline	1.03 ± 1.14	0.95 ± 1.32	0.747
3 months	0.90 ± 0.80	0.83 ± 1.06	0.716
6 months	0.89 ± 0.84	0.76 ± 0.85	0.413
Fats and oils			
Baseline	6.50 ± 2.45	6.19 ± 2.17	0.473
3 months	5.84 ± 2.13 ^a^	6.12 ± 2.04	0.489
6 months	5.72 ± 2.23 ^a,b^	5.96 ± 2.03	0.535

** *p* < 0.01, * 0.01 < *p* < 0.05; ^a^ Significant difference in difference between groups; ^b^ Significant difference in difference over a 6-month intervention period between groups in the repeated-measures ANOVA.

**Table 3 nutrients-13-03878-t003:** Multiple linear regression analysis between health coaching intervention, healthy diet behavior, and HbA1c.

	Model 1 ^a^ β(SE)	Model 2 β(SE)	Model 3 β(SE)	Model 4 β(SE)
Baseline HbA1c	−0.47 ** (0.08)	−0.45 ** (0.07)	−0.46 ** (0.07)	−0.44 ** (0.07)
Health coaching intervention	0.38 * (0.15)	0.26 (0.15)	0.32 * (0.15)	0.22 (0.15)
Whole grains intake		0.10 ** (0.03)		0.09 ** (0.03)
Fats and oils intake			0.12 * (0.05)	0.10 * (0.05)
R^2^	0.295	0.343	0.323	0.360

** *p* < 0.01, * 0.01 < *p* < 0.05; ^a^ Model 1: Pure coaching effect included only; Model 2 and Model 3: Detecting possible mediating effect between coaching effect and diet intake; Model 4: Final full model.

## Data Availability

The data presented in this study are available on request from the corresponding author. The data are not publicly available due to the Institutional Review Board of Cathay General Hospital privacy protection policy.

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
