# Peer review of "Effectiveness of Health Coaching in Diabetes Control and Lifestyle Improvement: A Randomized-Controlled Trial"

_nutrients, 2021, doi:10.3390/nu13113878_

Round 1
Reviewer 1 Report
In this study, Lin et al., describe effectiveness of a 6-month healthy diet coaching diet in the treatment of patients with type 2 diabetes. The study was conducted well in a randomized-controlled trial on 114 diabetic patients. However authors showed only HbA1 data which were significantly decreased. Authors need to include other metabolites data such as blood glucose levels, triglyceride and HDL and LDL levels, which will make the study more significant.
Author Response
Respectively reviewer:
For your valuable comments, our reply and article revision are as follows:
The comprehensive blood tests (including HDL, LDL, and triglyceride levels) provided for diabetic patients are done half-yearly according to our health regulations. Since this paper is designed and making use of the available data when patients make hospital visits every 3 months, it makes most of the blood test indicators unavailable for our follow-up tests, so the HbA1c level from the quarterly fast blood glucose tests is the most ideal indicator for this research. Moreover, our main study purpose is focused on blood glucose rather than other blood indicators, so we only use HbA1c as our main outcome indicator instead of incorporating other blood biochemical indicators such as HDL and LDL. However, in light of your recommendations, we have noted this in the Limitation section and explained why only HbA1c is included on this research paper at line 359-369.
We sincerely hope that our replies and revisions are acceptable to you. Thank you.

Reviewer 2 Report
Thank you for your interesting manuscript. However, I have some comments still for the improvement of it.
Major comments
- The objective of the study is missing. I understand that line 65-67, 70-71 seems to be objective although it did not describe as the objectives. In line 65-67, the authors mentioned the health diet and physical activities for looking at the effect of health coaching. However, I did not find the physical activity throughout the manuscript. Please recheck the objective of the study and its related matters.
- The one of the selection criteria for the participants was HbA1c of 7.0% or greater for the pass six month in line 97-98. If so, the participants with HbA1c between 6.5-6.9 (good control for diabetes) were not included in this study. Therefore, we should think about it in the analysis and discussion for missing this group of participants included in the study. Otherwise, you may have another reason for skipping those group.
- The authors reported the nutrition (serving/day) whole grains,…..,etc in the table 3 based on 2019 Taiwan dietary guidelines. You should describe the details of serving per day in the materials and methods section even though you refer the guidelines.
- In addition to comment 3, regarding with fat and oils you reported in the results, this reporting is very vague. Concerning with fats and oil, there are variety of fatty acids in Saturated, Monounsaturated and Polyunsaturated fatty acids, this might the different effect on HbA1c. Similarly, the different types of oils have different effect on diabetes directly or indirectly. Regarding with this, there are many research papers. For e.g https://www.mdpi.com/2072-6643/13/8/2625/pdf . Therefore, it is difficult to say the vague term “fats and oils” in the scientific paper, at least you have to identify the type of oils and fat. Otherwise, this will impact on the results as a bias, the validity of the study, as the weakness. In addition, I did not find how to remove this bias in the analysis.
- In the table 3, you described the multiple regression, did not differentiate as logistic or linear regression. Moreover, how to develop model 1, 2, 3, 4 is missing in both methods section and as the footnote under the table including how to fix the bias, confounding and mediators.
- The weakness of the study is need to be updated according to the above comments, also missing the strength of the study.
Minor comments
- The references are need to be updated throughout the manuscript, e.g the first reference is out dated now (from the data 2010). You can use the recent data from the International Diabetes Federation “Diabetes Atlas 9th edition 2019”.
- The introduction is need to fill with more information. E.g you should add the information regarding with the output variables such as nutrition.
- In line 97, the characteristic of the participants are described as the age between 20-75 years. On the other hand, the table 3 reported the data starting from the age of 30. You have to add the reason as the footnote under the table at least if it is not missing the data.
- The paragraph line 160-164 under the results does not relate the results. It is just only the methods.
Author Response
Respectively reviewer:
For your valuable comments, our reply and article revision are as follows:
Major comment replies:
Comment 1:
We would like to clarify that originally, we did have this item in our paper and were thinking about the publishers to submit to, and then upon receiving the invitation of a journal, we decided to leave out the ‘physical activity’ on the manuscript to better meet the requirement. This was due our part not properly revised.
Comment 2:
Based on your suggestion, we have added a paragraph to clarify this issue at line 300-314.
Comment 3:
We have added some recommendations of the guide to the Outcome section as well as the patients’ dietary adjustments suggested by dieticians and health coaches at line 160-168.
Comment 4:
In light of the lack distinction between different fats on both the guide and diabetes health education, we have made a note in the Limitation section at line 369-378.
Comment 5:
We have revised this to multiple linear regression and added footnotes to account for this modeling logic as according to your suggestion at line 249 and line 252-257.
Comment 6:
We have incorporated the strength of this study and added some limitations according to your suggestion at line 341-349.
Minor comment replies:
Comment 1:
According to your suggestion, we have amended this literature (the WHO 2016 version), checked the rest of the literature reference, and ensured that they are the latest data.
Comment 2:
We have made some adjustments according to your suggestion, making it clearer to elaborate the need to change the dietary composition at line 69-81.
Comment 3:
We have noted that no participants aged 20-29 were recruited according to your suggestion at line 203.
Comment 4:
We are not sure if you are referring to Line 160-164 in the original manuscript since they belong to Analysis Section not Result. Therefore, no revision is done for now.
We sincerely hope that our replies and revisions are acceptable to you. Thank you.

Round 2
Reviewer 2 Report
Thank you very much for your interesting manuscript. However, I have some suggestions for your manuscript.
Major comments
- The authors reported that healthy diet would be presented in two ways as daily calorie intake and nutrients in each servings. However, the daily calorie intake is missing in table 3. This means the study did not reach the objective?
- In line 159-162, the authors reported the recommended consumption for normal person for e.g 3-5 servings of vegetables. However, they missed to reported how much (grams) ? contains in each one serving. For example, the WHO recommended the one serving of vegetable contains around 80g of fruit and vegetables. Thus, you need to report it in details because healthy diet is your outcome variable.
- The description of statistical analysis in page 4 is not enough because the authors did miss the information regarding with table 3 (multiple linear regression) in details.
- In addition, in table 3, the authors reported the multivariate linear regression results via 4 models. However, they did not clearly mention how to build each model for detail information in all areas (under the title of statistical analysis, results and as the footnote under the table 3). The description is not enough statistically in the footnote of the table. Moreover, they did not report which model is the best.
- I understand that the health coaching intervention is the input variable and the HbA1C and healthy diet are outcome variables. However, the authors reported the health coaching intervention as also one of the outcome variable in table 3.
- You need to describe the sample size calculation formula you applied for RCT of this study.
Minor comments
- The first paragraph of the result section was related to the method section; it is not concerning the results. I already mentioned it in the first version.
Author Response
Respectively Reviewer:
Thank you for your comments, we have modified our manuscript, and listed our answer about your comments as below:
For Major comments:
- Since the total calorie intake had a high correlation with the whole grains intake (Pearson r=0.752, p< 0.001) and the fats and oils intake (Pearson r=0.596, p< 0.001), it could not add into regression at the same time. We added this explanation in lines 260-266 without creating one more model finally. We had tested and replaced the whole grains intake and fats and oils intake with the total calorie intake, and it produced a similar result as model 4, indicating that baseline A1c and total calorie intake were significant, and R2 was 0.333.
- Since the recommended weight varies among various types of food, our country’s nutrition and health education manuals mainly use the grams of nutrient components to calculate and provide reference weight guides for various foods rather than the weight of specific foods. For example, a serving of whole grains is equal to 80 grams of rice, 340 grams of corn, a piece of 120 grams of toast, etc. Therefore, we can only add the number of grams of nutrients to the text instead (in line 167-171); it’s impossible list the recommended weight of each item.
- The information about table 3 was originally located in lines 189-191. After careful evaluation, we decided to move the original footnote of Table 3 here and add a note to explain why the daily calorie intake was not included (in line 191-201). We believe that the revised description should be clear enough.
- Based on your suggestion, we have modified the description of the statistical analysis paragraph to make it clearer to explain the logic analysis of Table 3. We pointed out in the original footnote to Table 3 that Model 4 is the final full model, which means it is the model that finally incorporates all the variables, after considering the variables that should or should not be included. This should be what you mean by "the best one."
-
In Table 3, health coach intervention and healthy eating behavior are independent variables, and only HbA1c is the outcome variable. We have confirmed that whether it is the statistical method (lines 197-201) or the description in Table 3, health coaching intervention is never considered a result variable.
In this study, healthy eating behavior and HbA1c are both the result of health coach intervention, and the relationship between the three variables is found in Table 3 that the effect of health coach intervention on decreased HbA1c is completely mediated by the intervention effect of healthy eating behavior. - In our origin manuscript, it was described in lines 121-126. We have added another sentence to indicate that this is calculated by the statistical software GPower 3.1, so we cannot directly provide you the formula. We have also reviewed the description of sample size calculations in similar studies, so we believe that this paragraph of narrative should provide sufficient information.
For minor comments:
We have moved it to lines 97-101 of the method section according your suggestion.
